# Host kinase regulation of *Plasmodium vivax* dormant and replicating liver stages

**Elizabeth K. K. Glennon[1], Ling Wei[1], Wanlapa Roobsoong[2], Veronica I. Primavera[1], Elizabeth M. van Zyl[1], Tinotenda Tongogara[1], Conrad B. Yee[1], Jetsumon Sattabongkot[2], Alexis Kaushansky** [1,3,4]\*

**1** Center for Global Infectious Disease Research, Seattle Children's Research Institute, Seattle, Washington, United States of America, **2** Mahidol Vivax Research Unit, Mahidol University, Bangkok, Thailand, **3** Department of Pediatrics, University of Washington, Seattle, Washington, United States of America, **4** Department of Global Health, University of Washington, Seattle, Washington, United States of America

\* alexis.kaushansky@seattlechildrens.org

## Abstract

Upon transmission to the liver, *Plasmodium vivax* parasites form replicating schizonts, which progress to initiate blood-stage infection, or dormant hypnozoites that reactivate weeks to months after initial infection. *P. vivax* phenotypes in the field vary significantly, including the time to, and frequency of, relapse. Current evidence suggests that both parasite genetics and environmental factors underly this heterogeneity. Here, we applied an approach called kinase regression to evaluate the extent to which *P. vivax* liver-stage parasites are susceptible to changes in host kinase activity. We identified a role for a subset of host kinases in regulating the numbers of schizonts and hypnozoites, as well as schizont size, and characterized overlap as well as variability in host phosphosignaling dependencies between parasite forms across multiple patient isolates. Our data point to variability in host dependencies across *P. vivax* isolates, suggesting one possible origin of the heterogeneity observed in the field.

## Author summary

One of the major hurdles to malaria eradication is the ability of *Plasmodium vivax* to form dormant liver-stages called hypnozoites, which reside within the liver for weeks to months before being reactivated and causing a new round of blood-stage infection. Due to difficulties in culturing the parasite, little is known about the biology of *P. vivax* liver stage. Using an approach that combines a small chemical screen and machine learning, we investigate the host kinase dependencies of both dormant and developing liver-stage parasites across multiple *P. vivax* field isolates. We identify some overlap in host kinase regulation of parasite forms but observe a great deal of variation between isolates, suggesting parasite heterogeneity needs to be taken into consideration in future studies focused on host-parasite interactions within the liver.

**Data availability statement:** All data used in this submission are within the manuscript and supplemental files.

**Funding:** Parts of this work were funded by R21AI151344 and R01AI177257 from the National Institutes of Health (https://www.nih.gov/) to AK and EKKG. Funders did not play any role in the study design, data collection and analysis, decision to publish, or preparation of the manuscript.

**Competing interests:** The authors have declared that no competing interests exist.

## Introduction

Malaria is a significant global health burden with nearly 250 million clinical cases and more than a half million deaths annually world-wide [1]. Although significant progress has been made towards reducing the incidence of malaria, the *Plasmodium* parasite has developed resistance to all drugs in widespread use [2–4]. To initiate symptomatic blood-stage infection after transmission by a female *Anopheles* mosquito, the malaria parasite must first develop and replicate within the liver. Upon invading a hepatocyte, *Plasmodium* sporozoites develop as schizonts, growing and replicating to produce merozoites, which go on to invade red blood cells. A major hurdle to malaria eradication is the ability of *Plasmodium vivax* to form liver-resident dormant stages, called hypnozoites, in addition to the rapidly replicating schizont form [5]. Unlike schizonts, hypnozoites do not initially replicate their genomes, and instead reactivate weeks to months after the initial infection.

Although no DNA replication occurs, hypnozoites are not quiescent. Hypnozoites exhibit a modest increase in size over the course of infection both *in vitro* and *in vivo* and exhibit changes in gene expression, membrane protein distribution, and exhibit altered sensitivities to schizonticidal drugs over time [6–8]. Transcriptomic studies of *P. vivax* and *Plasmodium cynomolgi* dormant forms revealed enrichment of fatty acid synthesis and redox metabolism [7,9,10] suggesting that, despite their overall reduced transcriptional activity, hypnozoites are actively maintaining their redox state and may be susceptible to its perturbation.

Current data points to the model where a combination of parasite genetics and the host environment influences both maintenance and reactivation of *P. vivax* liver-stage (LS) parasites [11–13]. Given the potential for diversity in both parasite-intrinsic and environmental stimuli, it is not surprising that the phenotypes observed for *P. vivax* in the field are heterogeneous. The timing of relapse after hypnozoite activation has been shown to exhibit both periodicity and random constant rate patterns, suggesting that there are both triggered and stochastic processes that mediate the conversion between dormant and activated states [13]. Epidemiological studies suggest the rate of relapse varies ecogeographically [11,14]. It has also been shown that inflammation, particularly that elicited by blood-stage malaria (including of other species, such as *Plasmodium falciparum*), can trigger hypnozoite reactivation [12,15–17]. More recent models of *P. vivax* recurrence support roles for population and temporal heterogeneity in determining overall relapse patterns [18]. *P. vivax* isolates are traditionally divided into two variants, VK247 and VK210, based on the amino acid sequence of circumsporozoite protein (CSP). These variants differ in their mosquito vectors and have been shown to produce different ratios of hypnozoites to schizonts in a humanized mouse model [6,19]. However, the genetic diversity of *P. vivax* populations is not fully captured by this basic genotyping [20]. One longitudinal study identified 14 different *Pv*CSP alleles in patients at the Thai-Myanmar border [21] while another identified 67 unique *P. vivax* merozoite surface protein 1 (*Pv*MSP-1) haplotypes [22]. A large proportion of recurring *P. vivax* cases are heterologous infections and contain different parasite genotypes than acute infection [23], with a particularly high level of genetic diversity in parasites circulating in and around Thailand [23,24]. *In vitro* work

has also demonstrated a role for both parasite genetics and host environmental factors in regulating *P. vivax* infection, showing variation in infection rate, hypnozoite:schizont ratio, and schizont size across both parasite isolates and primary hepatocyte donors [25].

Work on schizont LS infection with *P. falciparum, P. yoelii, and P. berghei* has demonstrated parasite dependence on a multitude of host cell processes including the blocking of apoptosis [26–29], regulation of lipid peroxidation [30], nutrient uptake [31,32], and re-orientation of vesicular trafficking to the parasitophorous vacuole [33–36]. However, it is very unusual for individual studies to compare the reliance on host regulators of infection between *Plasmodium* species, so we are limited in our ability to extrapolate among them. Where direct comparisons among species have been conducted, variation, as well as similarities, in host factor dependencies has been uncovered [29,37–39]. *P. vivax* hypnozoites and schizonts have differing susceptibilities to many parasite-targeted drugs, but it remains an open question whether these parasite forms have common dependency on host determinants of infection. Inhibition of the water and small molecule channel aquaporin 3, which reduced LS *P. berghei* [32], also inhibited *P. vivax* schizonts and hypnozoites [8]. *P. vivax* schizonts and hypnozoites exhibited overlapping susceptibility to treatment with interferons (IFN), with both IFNγ and IFNα reducing hypnozoite and schizont levels *in vitro* [40]. While several broad genetic screens have been conducted to identify host genes that influence LS infection with other, more tractable, *Plasmodium* species [33,34,41], the relative intractability of primary hepatocytes to genetic manipulation and the difficulty of producing *P. vivax* sporozoites, make it particularly challenging to conduct large genetic screens. Currently, the full extent to which *P. vivax* LS infection is susceptible to manipulation of the host environment remains an open question.

To more broadly probe the host dependencies of *P. vivax* LS infection we have adapted kinase regression (KiR), which utilizes the characterized and quantified polypharmacology of kinase inhibitors and machine learning to predict which of 300 human kinases play a role in a cellular phenotype. In KiR, a set of ~30 commercially available compounds with known activities against each of ~300 human kinases [42] are tested for their capacity to inhibit any quantitative phenotype. Using machine learning, these two data sets are combined to generate predictions of kinases that contribute most substantially to the given cellular outcome. This approach was first pioneered to identify regulators of cell migration [43], and then determinants of cancer susceptibility [44–46], Kaposi's sarcoma-associated herpesvirus reactivation [47], and SARS-CoV-2 induced cytokine release [48]. We have utilized KiR to predict both previously described and novel host kinase regulators of *P. yoelii* LS infection [49] as well as to identify kinases that regulate the blood-brain endothelial barrier [50–52]. We hypothesized that the host environment plays a major role in regulating *P. vivax* LS biology and have utilized KiR to probe the extent to which hypnozoites and schizonts are dependent on host phosphosignaling across multiple parasite isolates.

## Results

### *P. vivax* isolates exhibit different infection phenotypes in primary human hepatocytes

To probe the role of host kinase signaling in regulating *P. vivax* liver-stage (LS) infection we utilized the 384-well primary human hepatocyte microculture system described previously [53]. To reduce the number of confounding variables, the same lot of primary human hepatocytes was used for all experiments in this paper. We infected hepatocytes at an MOI of ~0.5 with sporozoites derived from four separate clinical isolates, spread across two independent infections. We termed the four clinical isolates A, B, C, and D; infections from isolates B and C were performed at the same time. Parasite isolates were collected in the Tak province in northwestern Thailand and were all the VK210 CSP variant, which is highly prevalent in Thailand [54,55]. 60,000 sporozoites were isolated per mosquito for isolate B, 64,000 per mosquito for isolate C, and 45,000 per mosquito for isolate D. Counts per mosquito were not recorded for isolate A. We allowed the parasites to develop for eight days post-infection and quantified hypnozoites and schizonts by microscopy. Liver-stage parasites were identified by circumferential *P. vivax* Upregulated in Infectious Sporozoites-4 (*Pv*UIS4) staining. Schizonts were distinguished from hypnozoites by additional morphological features: schizonts were defined as having multiple nuclear

masses and a diameter of over 10µm, and hypnozoites were defined as having a diameter of under 8µm and displaying the characteristic UIS4 prominence (**Fig 1a**) [6,56]. Even in the absence of any treatment, parasites displayed significantly varying phenotypes between the four isolates used. Infection rates were significantly lower in isolate C, with hypnozoite levels below robustly quantifiable levels (**Fig 1b**, **1c**). Isolates A, B, and D had comparable overall infection rates but significantly different ratios of hypnozoites to schizonts (**Fig 1b**, **1c**). Schizont size spanned a large range and was significantly higher in isolates C and D compared to the other two isolates (**Fig 1d**). The variance of schizont size was also significantly higher in isolate C compared to isolate A.

## Host kinase inhibition influences schizont and hypnozoite infection levels

We have previously demonstrated that, using a panel of 34 kinase inhibitors (KIs), novel kinase regulators of *Plasmodium* LS infection can be uncovered by KiR [49]. To apply KiR to *P. vivax* infection, we infected primary hepatocytes with each of three isolates, A, B and C, and treated each infected culture with the panel of 34 inhibitors beginning 24 hours post-infection (hpi) (**S1 Table**). Isolate D was used to test predictions. Inhibitors previously shown to reduce *P. falciparum* blood-stage (BS) infection [49] were excluded from the screen, as we reasoned that they are more likely to be targeting *Plasmodium* kinases directly, making their activity difficult to interpret. One inhibitor, staurosporine (KI29), that exhibited significant toxicity over 20% in primary hepatocytes (**S1 Fig**), was also removed from the screen. No other inhibitors exhibited statistically significant toxicity >20% across all three experiments. Schizonts and hypnozoites were quantified in response to treatment with the panel of inhibitors (**Fig 2**). Edge effects were tested for by comparing schizont and hypnozoite levels in technical replicates in outer edge or inner wells for each biological replicate. No significant effect of well position on infection was observed between paired wells that received the same treatment (**S2 Fig**). In parallel experiments conducted with the same parasite isolates, treatment with the phosphatidylinositol 4-kinase inhibitor, MMV390048, significantly reduced schizont numbers, as expected (**Fig 2**, data published in [57]). Kinase inhibitors exhibited a range of effects across forms and isolates (**Fig 2**). Analysis of variance revealed kinase inhibitor treatment led to a significant difference in schizont and hypnozoite numbers than would be expected from random sampling, for all parasite strains and forms (**Fig 2**). Strikingly, many inhibitors exhibited different effects across the different isolates. However, some inhibitors did exhibit a similar effect across isolates. Specifically, two inhibitors, GSK-3 inhibitor X (KI15) and K252a (KI20)

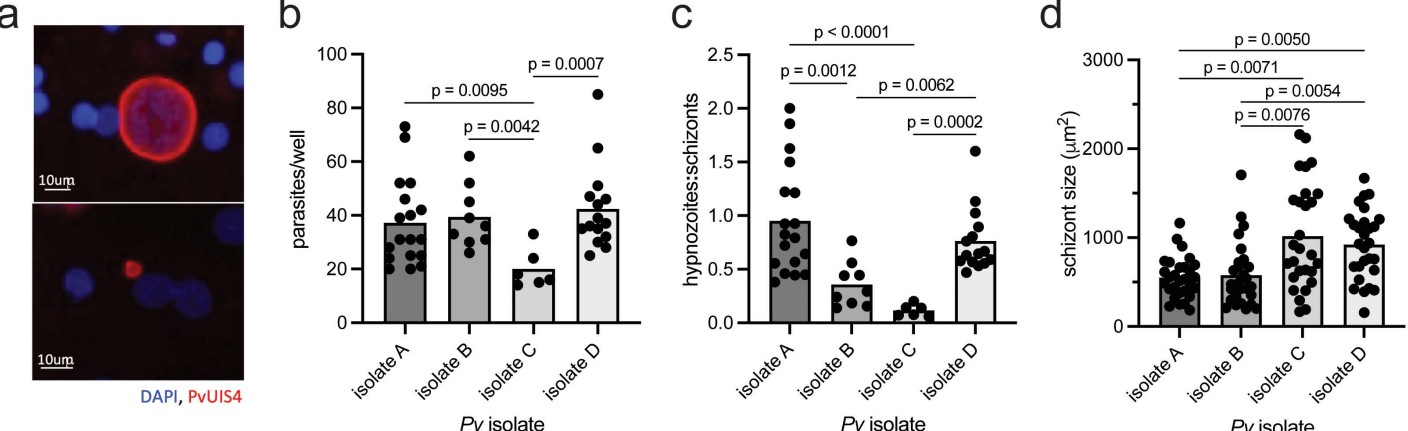

**Fig 1. *P. vivax* parasites exhibit variation across different patient isolates.** (a) Representative images of *P. vivax* (*Pv*) schizont (top) and hypnozoite (bottom) 8 dpi in primary human hepatocytes stained with DAPI and *Pv* Upregulated in Infective Sporozoites 4 (*Pv*UIS4). Comparison of (b) parasite numbers, (c) ratio of parasite forms, and (d) schizont size across four unique *P. vivax* isolates 8 days post-infection. Each dot represents counts from a control well. Data were analyzed by Kruskal-Wallis test with Dunn's uncorrected multiple comparison test.

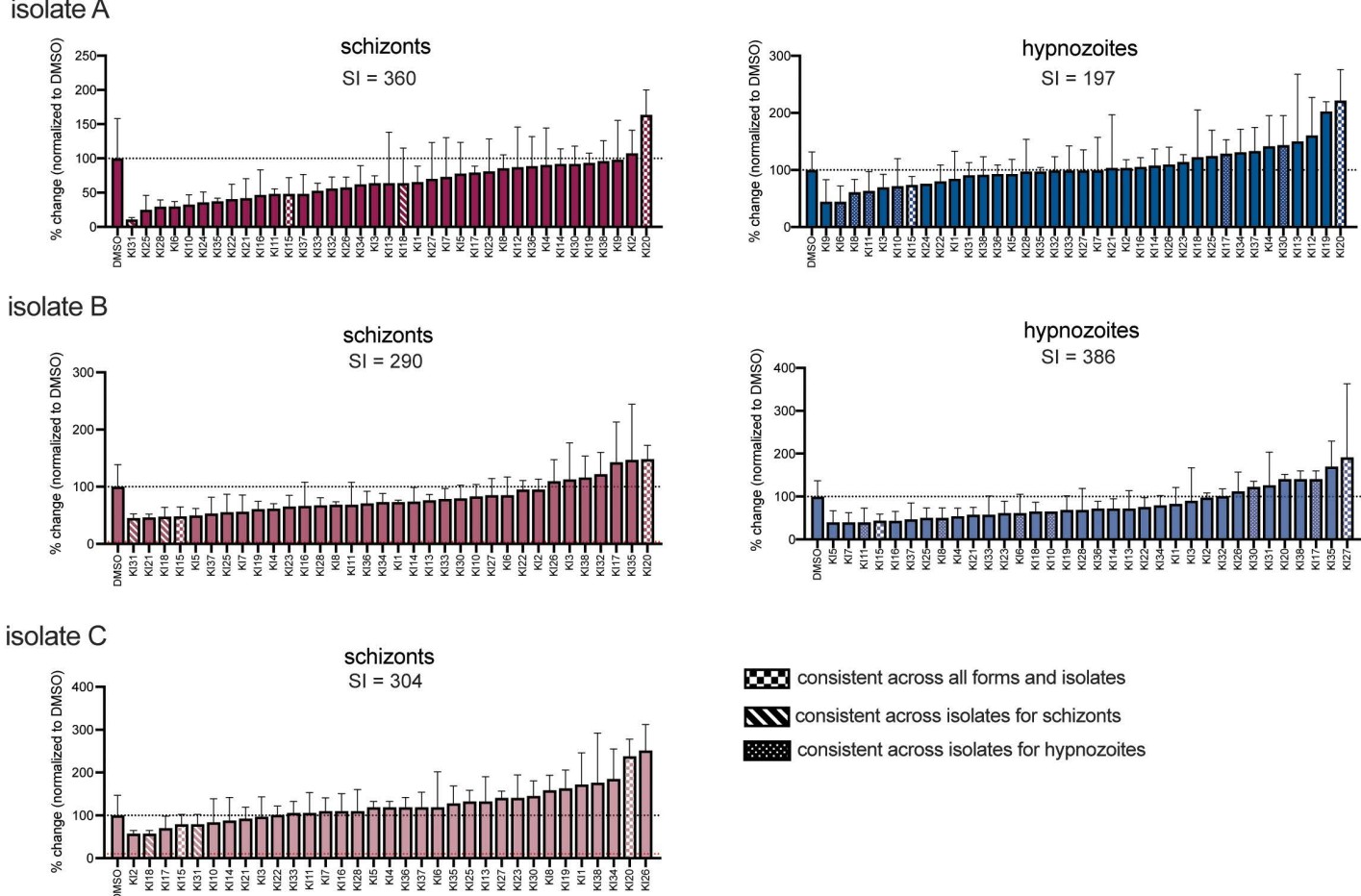

**Fig 2. Kinase inhibitors alter numbers of *P. vivax* schizonts and hypnozoites.** Effect of a panel of 28 kinase inhibitors on infection levels of three different *P. vivax* isolates in the same lot of primary human hepatocytes. All inhibitors were used at 500nM. Schizonts and hypnozoites were quantified 8 dpi from 3 technical replicates for each inhibitor and normalized to a DMSO control. Data are shown as mean and standard deviation of 3 technical replicates for each treatment. Inhibitors that consistently increased or decreased parasite numbers across forms and/or isolates, using a 20% change cut-off, are indicated. A sensitivity index (SI) was calculated for each form and isolate. Graphs are color-coded by parasite form (schizonts in pink, hypnozoites in blue) and by isolate (color intensity). Red dashed line indicates % change measured in response to 20 µM MMV390048 treatment measured in parallel experiments [57]. Data were analyzed by one-way ANOVA.

consistently reduced and increased the number of schizonts and hypnozoites, respectively, across all isolates, using a 20% change cutoff. Two additional KIs (KI31 and KI18) consistently reduced schizont numbers across all isolates. Four additional KIs (KI6, KI8, KI10, KI11) consistently reduced, and two additional KIs (KI17 and KI30) consistently increased, the number of hypnozoites.

We next asked if the number of schizonts was more likely to be altered by kinase inhibitor treatment, overall, than the number of hypnozoites. However, a strict average of inhibition does not provide a robust measure of overall sensitivity to host phosphosignaling inhibition because kinase inhibitors within the panel vary tremendously in their level of promiscuity. The number of kinases inhibited by a single KI by >30% ranged from 1 to 228 within the panel of 300 kinases for which we have published data [42]. Within the panel of KIs, more promiscuous inhibitors did not necessarily inhibit infection by a greater extent. Comparing the number of kinases inhibited by each individual inhibitor with its effect on schizont or hypnozoite numbers, for isolate A, showed a moderate positive correlation across a range of inhibition cut-off levels from

20-80%, with the most promiscuous KI in the screen, K252a (KI20), inducing the largest increase in both schizonts and hypnozoites (**S3 Fig**). To rigorously evaluate how the overall phosphosignaling landscape impacts schizont and hypnozoite levels, we developed a "sensitivity index" (SI), which accounts for the polypharmacology of each kinase inhibitor by assigning each inhibitor a weight based on the extent of inhibition across the 300 kinases in the previously published dataset (**S1 Table**) [42]. These weights were then integrated with the effect of each inhibitor on schizont or hypnozoite numbers to calculate an overall measure of sensitivity to kinase inhibition for each phenotype. Using this statistic, we observed variation in sensitivity both between forms and parasite isolates, with hypnozoite levels for some isolates showing more sensitivity to manipulation of host signaling than their schizont counterparts (**Fig 2**). This was surprising, given the strong trend of hypnozoites not responding to even very effective anti-schizont therapies.

The ratio of hypnozoites to schizonts formed upon infection is known to be influenced by parasite genetics [11,14]. However, there is also evidence that the host environment can influence hypnozoite:schizont ratio [25] and reactivation [12,15–17]. We asked if our data were consistent with the hypothesis that a reduction in schizonts or hypnozoites upon KI treatment was due to a shift in forms: more schizonts corresponding with fewer hypnozoites or vice versa. If this hypothesis was true, we would expect to observe an inverse correlation between the impact on schizont and hypnozoite numbers across treatments. In fact, we observe a strong positive correlation between the impact of a given inhibitor on schizonts and hypnozoites. This suggests that drugs that inhibit schizonts are also likely to inhibit hypnozoites. The underlying biological origins of this phenomenon remain unknown, but it does suggest that the targets of the most effective inhibitors are relevant for both schizont growth and hypnozoite survival. (**Fig 3**). Using a 20% change cut-off, two KIs were identified in each of two isolates that increased hypnozoite numbers while decreasing schizont numbers, however which KIs had this effect was not conserved between isolates. No KIs decreased hypnozoite numbers while also increasing schizont numbers.

## Using kinase regression to predict susceptibility of *P. vivax* liver-stages to kinase inhibitors

Kinase regression (KiR) utilizes the overlapping specificities of a small panel of kinase inhibitors to deconvolve the effects of 300 host kinases, and over 100 additional kinase inhibitors not in the screen, on a quantifiable phenotype (**Fig 4a**) [43].

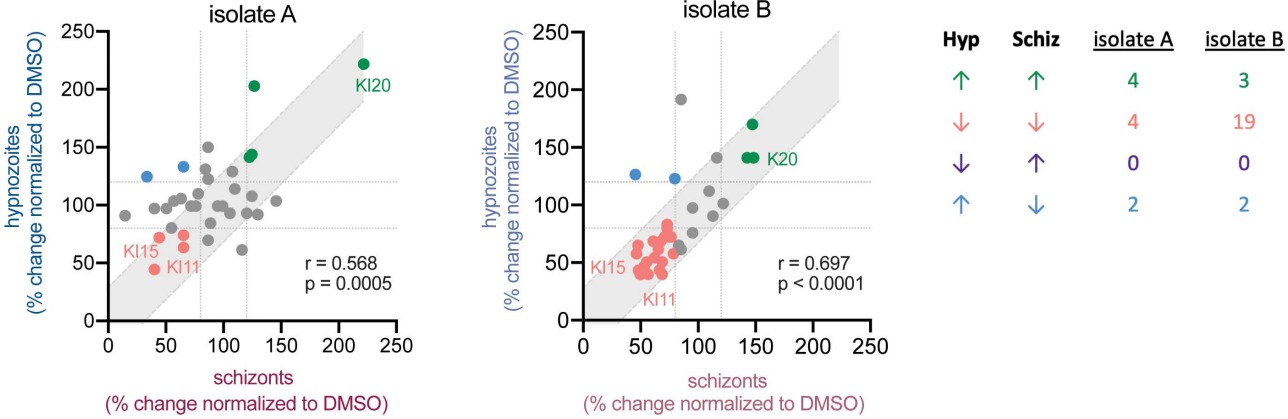

**Fig 3. Effect of KIs on hypnozoites and schizonts correlates within each isolate.** Effect of each kinase inhibitor (KI) on number of schizonts (Schiz) plotted against its effect on number of hypnozoites (Hyp) for each isolate. Each dot represents the mean % change of 3 technical replicates compared to DMSO controls for a single kinase inhibitor. Gray shaded area indicates a 95% confidence interval, based on the variation of control wells, around x = y. A 20% change cut-off, indicated by the dotted gray lines, was used to categorize the effect of inhibitors on each parasite form. Inhibitors are color coded based on whether they reduced numbers of both forms, increased numbers of both forms, or effected only one form. Inhibitors that consistently fell into the same quadrant are labeled. The number of inhibitors that fell into each category for each isolate are summarized in the table. Inhibitors that had a consistent effect across both isolates are labeled. Pearson correlation coefficient (r) and p-value (p) are indicated for each graph.

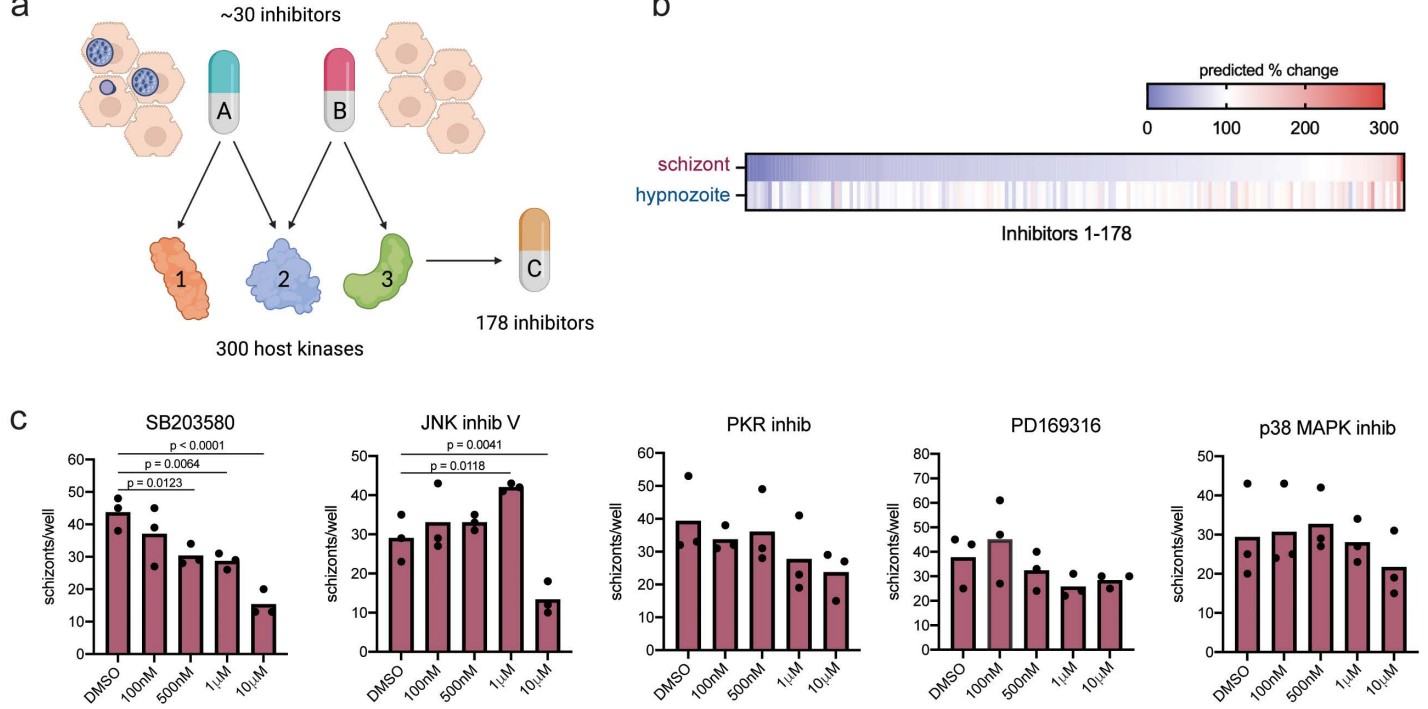

**Fig 4. Kinase regression predicts kinase inhibitors that alter *P. vivax* infection. (a)** Schematic of kinase regression (Created in BioRender. Glennon, E. (2025) https://BioRender.com/ u2nzz5c). Briefly, the effect a panel of ~30 inhibitors, with characterized overlapping kinase specificities, on *P. vivax* liver-stage (LS) infection was measured. These data were then used to predict the role of 300 host kinases and the effect of 178 kinase inhibitors on LS infection. **(b)** Heatmap showing predicted effect, as percent change compared to DMSO controls, of 178 inhibitors on the number of parasites based on the screen conducted in isolate A (Fig 2). Inhibitors are ordered based on magnitude of predicted effect on schizont numbers **(c)** Schizont dose response curves in isolate B for five inhibitors that were predicted to reduce infection based on screen conducted in isolate A. Primary human hepatocytes were infected with *P. vivax* sporozoites from isolate B and treated beginning 24hpi post through 8 dpi when schizonts were quantified. Each dot represents a technical replicate. Graphs are color-coded by isolate and by form. Data were analyzed by Fisher's LSD test.

We previously used KiR to identify kinases and kinase inhibitors that altered *P. yoelii* infection, with 85% accuracy [49]. Because each *P. vivax* parasite isolate originated from an independent patient and the impact of inhibitors varied across isolates (Fig 2) we used KiR to make predictions for each isolate individually (S2 Table). We next wanted to test the accuracy of these predictions using an independent patient isolate. To do this, we used the data from isolate A to predict the effect of 178 inhibitors on the number of each parasite form (Fig 4b), and selected five that were predicted to reduce schizont levels by > 80%, based on data from isolate A, for testing in isolate B. Primary hepatocytes infected with parasites from isolate B were treated with the 5 inhibitors from 24 hpi through 8 dpi when schizonts were quantified (Fig 4c). Consistent with the variation in the response of different parasite isolates to kinase inhibition, only one of the five inhibitors, SB203580, significantly reduced schizont numbers with a dose response in isolate B (Fig 4c). As an internal check of the predictive power of KiR we also compared the level of schizont inhibition of the 5 selected inhibitors (Fig 4c) to their predicted inhibition levels based on screen data from the same isolate. All 5 inhibitors were predicted to reduce schizont numbers in isolate B, with predicted levels falling between 21–70% of controls (S2 Table). There was no significant correlation between predicted and mean tested levels at any of the concentrations tested (S3 Table), suggesting predictions of inhibition magnitude by KiR may not be robust, possibly due to differences in inhibitor bioavailability.

To further investigate differences in inhibitor sensitivity between parasite isolates, we conducted dose response curves in isolate B for two inhibitors that were part of the original screen: SU11274 (KI31), which reduced schizont numbers

across isolates A and B, and casein kinase I inhibitor D4476 (KI6), which reduced the number of schizonts in isolate A but not isolate B (**S4a Fig**). KI31 (SU11274) reduced schizont numbers with increasing magnitude as concentration increased (**S4b Fig**). KI6 (casein kinase I inhibitor D4476), which did not significantly impact schizont numbers in isolate B in the screen (500nM), only reduced the number of schizonts at the highest concentration tested (10µM) (**S4b Fig**).

To more broadly probe the predictive power between isolates for susceptibility to host kinase inhibition we used the KI screen data from one isolate to predict drug susceptibility and then compared those predictions to the KI screen data from another isolate (see Methods). Predictions and screen data did not correlate strongly between separate isolates. Only one isolate pair showed a statistically significant weak correlation for schizont levels (**Fig 5a**) while no correlation was observed for hypnozoite numbers from separate isolates (**Fig 5b**). As we had observed some consistency in the effects of KIs on hypnozoites and schizonts within an isolate (**Figs 2**, **3**) we asked if data from one form could predict activity for

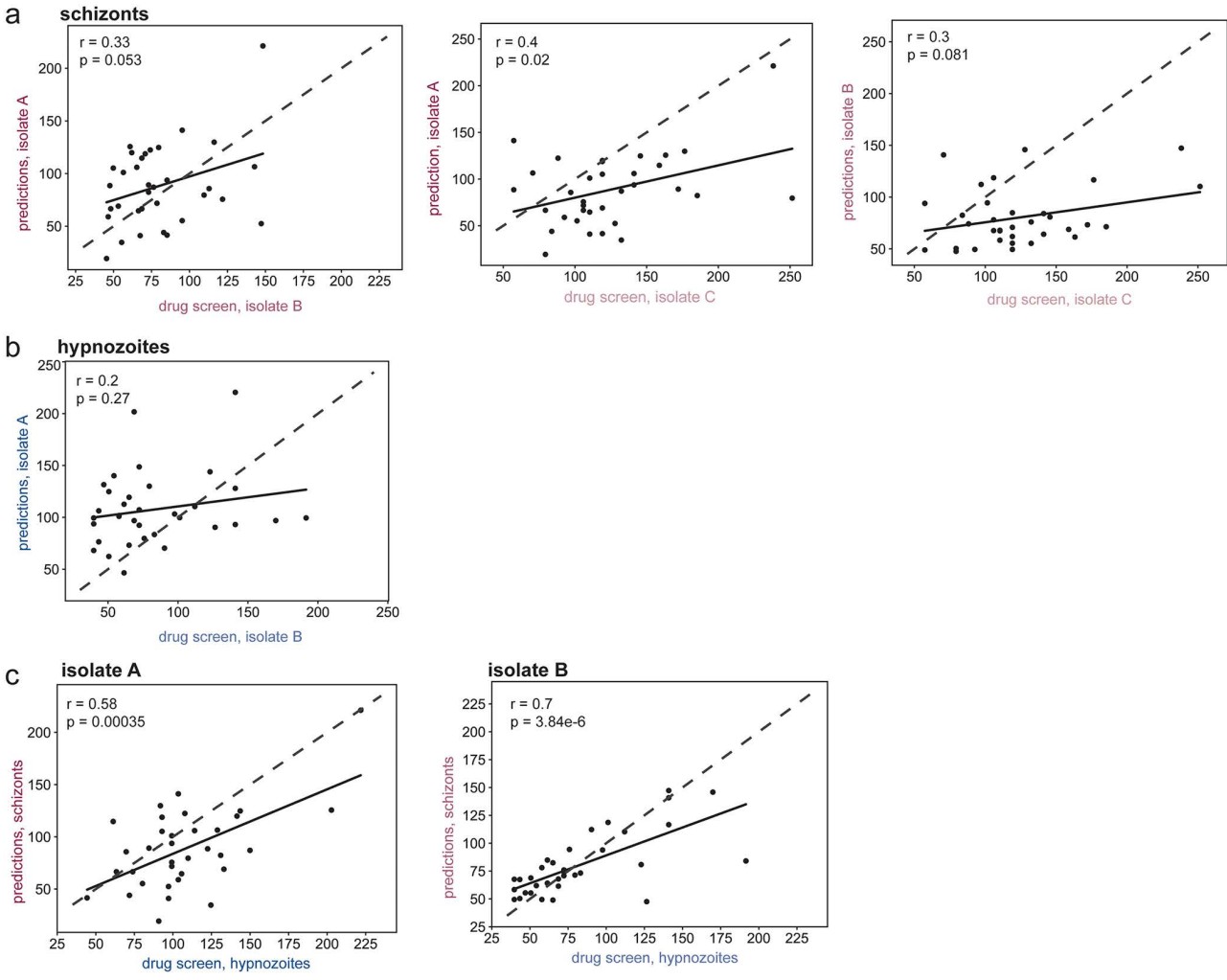

**Fig 5. Predictive power of KiR across parasite isolates and forms.** KI effect on **(a)** schizont or **(b)** hypnozoite numbers was predicted using data from one isolate and graphed against drug screen data from another isolate. **(c)** KI effect on infection was predicted using schizont data and graphed against drug screen data for hypnozoites within the same parasite isolate. Data are shown as percent change compared to DMSO controls at 8 dpi. Each dot represents a single inhibitor. Pearson correlation coefficients (r) were calculated for each comparison. P-values (p) are shown for each correlation.

the other form within the same isolate. Kinase inhibitor susceptibility predictions made using screen data from schizonts correlated strongly with the drug screen data for hypnozoites within each isolate (**Fig 5c**), suggesting isolate identity may play a stronger role than parasite stage in determining *P. vivax* LS susceptibility to host kinase perturbation.

Finally, we tried to identify kinase inhibitors with conserved effects across isolates A-C for testing in a fourth isolate (isolate D), which exhibited a similar total infection rate to isolates A and B, and a similar hypnozoite to schizont ratio to isolate A (**Fig 1b**, **1c**). Five new inhibitors were selected that showed some level of inhibition across isolates A-C. Specifically, we selected inhibitors that were predicted to reduce schizont levels by greater than 40% in two or more isolates and hypnozoites in at least one isolate (**S4 Table**). Infected hepatocytes were treated with each inhibitor at concentrations between 10nM and 10μM from 24 hpi to 8 dpi when cells were fixed and parasites quantified. All five inhibitors significantly reduced schizont numbers at one of the concentrations tested, and four of five significantly reduced hypnozoite numbers at the highest concentration tested (**Fig 6a**, **6b**). This was a much higher success rate compared to inhibitor predictions made from data on only one isolate (**Fig 4c**), suggesting that pooling data from several isolates can lead to robust predictions despite inter-isolate variation.

## Host kinase inhibition regulates schizont size

Multiple studies suggest that parasite-intrinsic and/or host factors might influence the rate of schizont growth. When comparing four hepatocyte donors, Vantaux and colleagues demonstrated that *P. vivax* LS schizonts can vary dramatically in size across different hepatocyte donors [25], although they observed modest differences in schizont size when comparing isolates. In contrast, we observed significant differences in both average and variance of schizont size between *P. vivax*

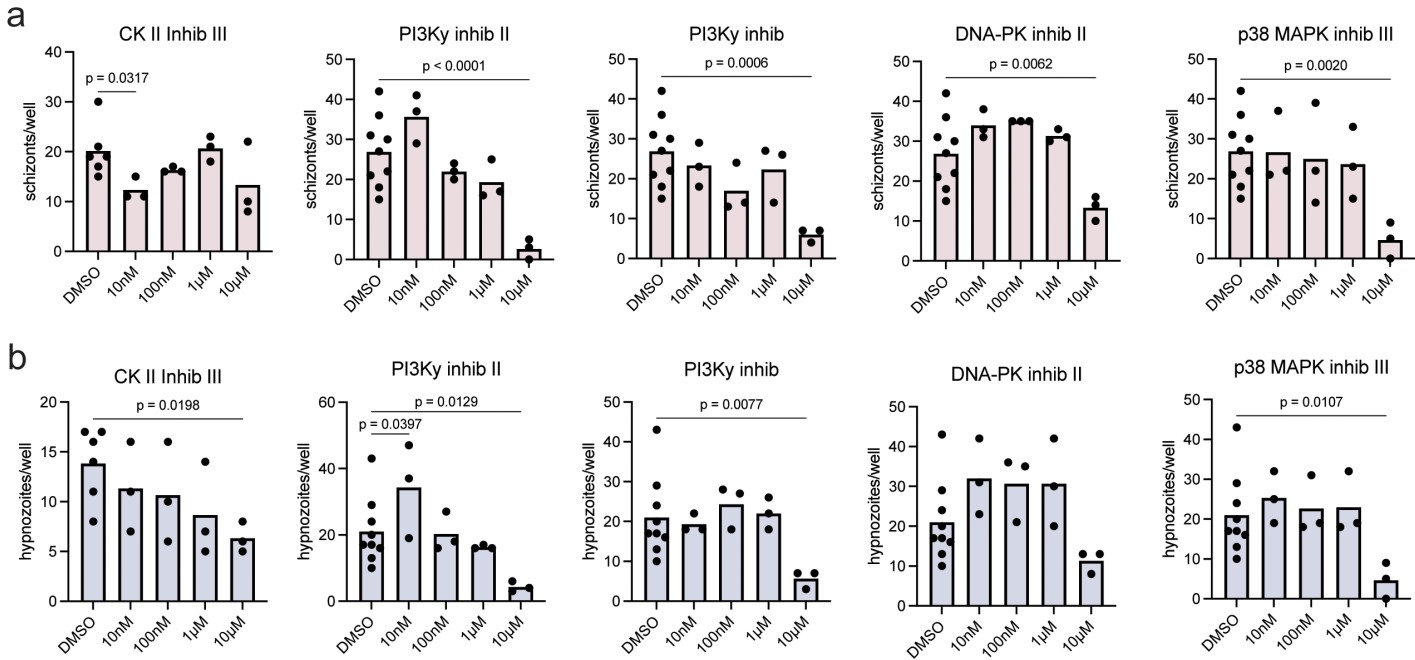

**Fig 6. Inhibitors predicted across multiple isolates reduce schizont and hypnozoite numbers.** Dose response curves for **(a)** schizont and **(b)** hypnozoite numbers in isolate D for five inhibitors that were predicted to reduce infection the most consistently across isolates A-C (**S4 Table**). Primary human hepatocytes were infected with *P. vivax* sporozoites from isolate B and treated beginning 24h post-invasion through 8 dpi when schizonts and hypnozoites were quantified. Each dot represents a technical replicate. Graphs are color-coded by isolate and by form. Data were analyzed by Fisher's LSD test.

isolates within the same hepatocyte donor (**Fig 1**). Parasite size likely serves as a surrogate for parasite biomass, as in untreated wells schizont size correlated strongly with the parasite nuclear stain signal (**S5a Fig**).

When we measured the effect of the kinase inhibitor panel on the size of schizonts, we again saw minimal overlap in effect between isolates (**Fig 7**). Two KIs consistently reduced schizont size, one of which (SU11274/KI31) had also consistently reduced the number of schizonts. K252a (KI20), which consistently increased the number of both parasite forms, also increased schizont size in all isolates. A SI was calculated for size for each isolate in the same manner as was done

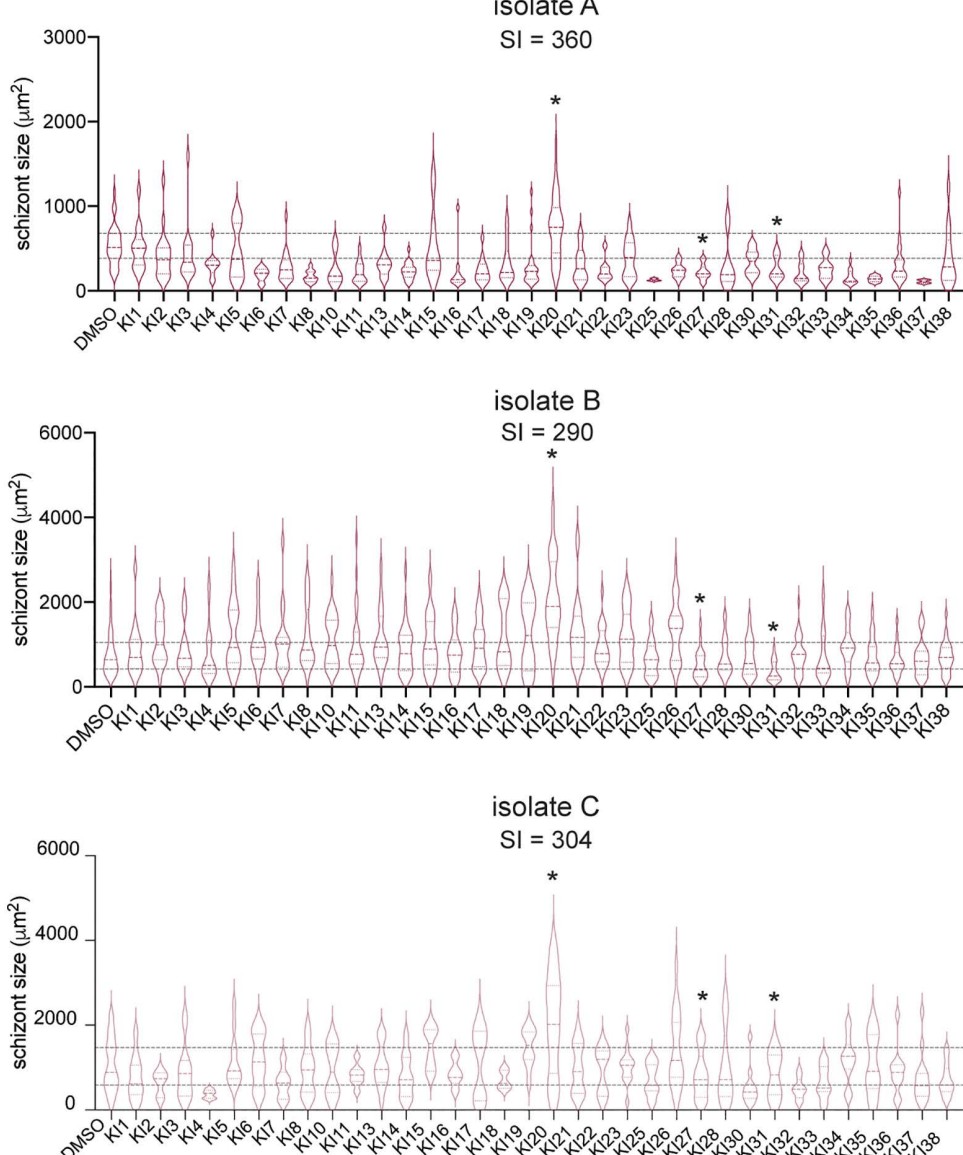

**Fig 7. Host kinase inhibition influences schizont size.** Effect of panel of 28 kinase inhibitors on schizont size for three different *P. vivax* isolates in primary human hepatocytes. Schizont area was quantified from 3 technical replicates for each inhibitor and normalized to a DMSO control 8 dpi. Inhibitors that consistently increased or decreased size across all 3 isolates, using a 20% cut-off, are highlighted. A sensitivity index (SI) was calculated for each isolate. Graphs are color-coded by isolate.

for parasite numbers (**Fig 7**). Using this metric, we again observed variation among isolates in sensitivity to kinase inhibition. The effect of KIs on schizont size did correlate between each isolate pair, however the strength of this correlation was driven by the effect of K252a (KI20) (**S5b Fig**). A significant correlation existed between the effect of KIs on schizont levels and the effect on schizont size for isolates A and C, but not for isolate B (**S5c Fig**). These data suggest that there may be distinct host factors that influence the presence or absence of parasites and parasite size.

## Host kinase regulation of *P. vivax* LS infection across isolates and phenotypes

We next asked if the KI screen could identify host regulators of *P. vivax* infection. We used KiR to predict host kinases whose activity impacts the number of schizonts, hypnozoites, and schizont size. Data from each isolate (**Figs 2**, **7**) were run through KiR independently and kinases identified that were predicted for at least 2 of the 3 isolates were considered bona fide predictions (**Fig 8a**, **S2 Table**). One kinase, ErbB2 receptor tyrosine kinase 2 (ERBB2), a member of the epidermal growth factor receptor (EGFR) family of receptor tyrosine kinases and an oncogene, was predicted to regulate all three phenotypes: schizont levels, hypnozoite levels, and schizont size across multiple isolates. Casein kinase 2 alpha 1 (CSNK2A1) was predicted to regulate the number of both schizonts and hypnozoites, however this could be confounded by a direct effect of inhibitors on the parasite casein kinase [58,59]. Activin A receptor type 1B (ACVR1B), a growth and

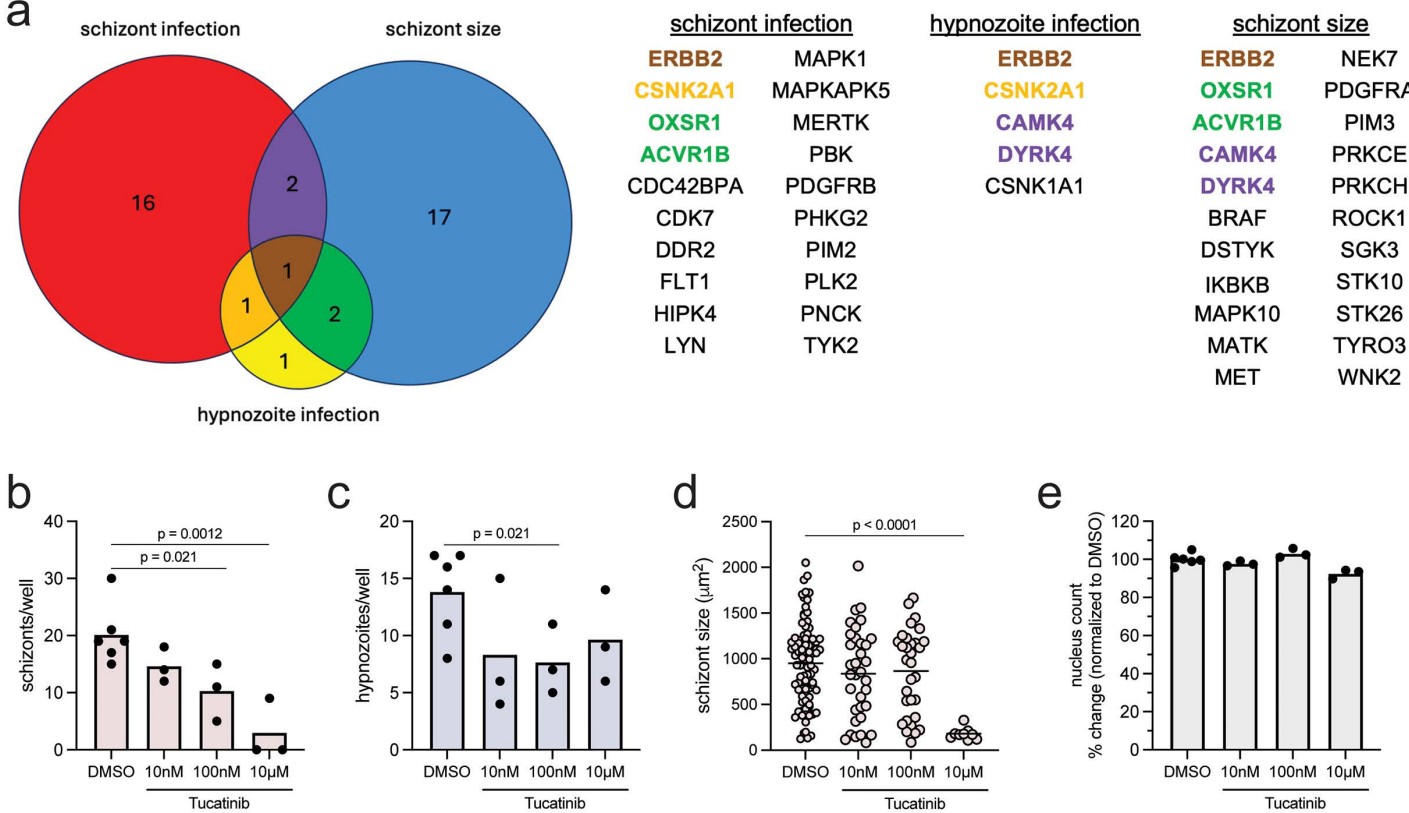

**Fig 8. Host kinases predicted to regulate *P. vivax* infection.** (a) Overlap in host kinases predicted to regulate schizont numbers, hypnozoite numbers, and schizont size in at least two of three isolates at alpha = 0.8. Kinases predicted to regulate more than one phenotype are indicated bold and color-coded. (b) Number of schizonts, (c) number of hypnozoites, and (d) schizont size 8 dpi in isolate D in response to treatment with tucatinib. (e) Hepatocyte nuclear counts per well normalized to the average of DMSO control wells. Treatment was begun 24 hpi and maintained until fixation. Each dot represents a technical replicate. Data were analyzed by ANOVA with Fisher's LSD test. * p ≤ 0.05, ** p < 0.01, *** p < 0.0001.

differentiation factor, and oxidative stress responsive kinase 1 (OXSR1), were predicted to regulate both the number and size of schizonts, consistent with previous work describing reactive oxygen species and lipid peroxidation regulation of *P. yoelii* infection and size [30,60]. Two kinases were predicted to regulate both schizont size and hypnozoite numbers: calcium/calmodulin-dependent protein kinase 4 (CAMK4) and dual specificity tyrosine phosphorylation regulated kinase 4 (DYRK4).

To test the prediction that host ERBB2 regulates *P. vivax* infection, as well as schizont size, we utilized the small molecule inhibitor tucatinib, which inhibits ERBB2 with high selectivity compared to over 200 other kinases screened, including other members of the EGFR family [61]. Two-hour pre-incubation with 10μM tucatinib reduced levels of ERK1/2 and AKT phosphorylation in response to EGF stimulation in uninfected cells (**S6a**-**S6b Fig**), as would be expected in the context of ERBB2 inhibition [62]. Primary human hepatocytes were infected with *P. vivax* isolate D (**Fig 1b**-**1d**) and isolate E (**S6c**-**S6e Fig**), for which we did not have kinase predictions, and treated with tucatinib at concentrations between 10nM to 10μM from 24 hpi until fixation. Isolate D was fixed and parasites quantified 8 dpi. The number of schizonts decreased in a dose-responsive manner upon tucatinib treatment (**Fig 8b**) while the number of hypnozoites decreased only modestly at 100nM (**Fig 8c**). Schizont size was only significantly reduced with 10μM treatment (**Fig 8d**). Due to concerns about contamination, parasites were quantified 5 dpi for isolate E. Contaminated wells were excluded from all analyses. We observed consistent effects of tucatinib on schizont numbers and size (**S6c**-**S6d Fig**), however we did not see an effect on hypnozoite numbers (**S6e Fig**). This could be due to late clearance of hypnozoites by tucatinib treatment or due to differences between the parasite isolates. No concentration of tucatinib significantly reduced host cell viability by more than 10% compared to controls (**Fig 8e**), suggesting ERBB2 inhibition selectively kills parasites, or infected cells, with minimal toxicity towards uninfected hepatocytes. Consistent with our results, Maher et al. recently published data demonstrating the potent anti-LS activity of another ERBB2 (HER2) inhibitor, poziotinib, against *P. vivax* and *P. cynomolgi* schizonts and hypnozoites [63].

## Discussion

*P. vivax* exhibits high levels of genetic and phenotypic diversity. Studies have identified up to 45% of patient isolates as genetically mixed and genetic diversity in local *P. vivax* populations is estimated to be higher than in the global *P. falciparum* population [64,65]. Infection rate and the frequency of hypnozoite reactivation are influenced by geographic region and by individual population heterogeneity [14,18]. Our data demonstrate a large amount of variation across *P. vivax* isolates in the hepatocyte phosphosignaling pathways that regulate their LS infection rates. Our data are consistent with the hypothesis that there is significant variation in the biology of *P. vivax* LS infection between individual parasite isolates collected from the same region, within the same CSP variant, and within the same source of primary hepatocytes. Consistent with published work using the VK210 variant *in vitro* [25] and in the FRG huHep mouse model [6], we found that parasite size, infection rate, and hypnozoite to schizont ratio varied significantly among parasite isolates under basal conditions, although we did observe more variation in schizont size among isolates within a single hepatocyte donor compared to Vantaux et al [25]. Furthermore, we observed a large amount variation among *P. vivax* isolates in both the extent and specificity of their sensitivity to host kinase inhibition. Two parasite isolates that were processed in parallel did not show less variation compared to each other than compared to the isolate processed on a different date, suggesting that biological difference, rather than technical experimental variation, is driving the observed phenotypes. Significant heterogeneity has been described in genotype [66,67], transcriptional profiles [40,68–71], and infection phenotypes across the *P. vivax* lifecycle. Varying infection phenotypes include disease severity [72,73], selectivity index for reticulocytes [74], and the frequency and timing of relapse [6,11,12,14,18]. Our data suggest a potential mechanism that could explain some of these heterogeneous outcomes within a single parasite variant; that differing responses to host phosphosignaling could underly the infection phenotypes of different *P. vivax* isolates. The two kinase inhibitors that consistently reduced (GSK3 inhib X - KI15) and increased (K252a - KI20) all 3 phenotypes across isolates A-C are both highly promiscuous inhibitors [42], so it

is possible that different mechanisms may underly their effect across the phenotypes and isolates tested. Whether these differences in host phosphosignaling susceptibility extend beyond the liver stage of infection is an interesting area for future investigations.

Within the LS, the greater consistency (Figs 2, 3) and predictive power (Fig 5) between schizont and hypnozoite forms within an isolate suggests that the genetic diversity between individual parasite isolates is a greater driver of phenotypic diversity in the response to host kinase inhibition than the differences between dormant and replicating parasite forms. Single cell transcriptional profiling of LS *P. vivax* and *P. cynomolgi* parasites identified multiple sub-clusters within schizont and hypnozoite populations [40,71,75]. Sub-clusters of schizonts were posited to be defined by their pre-commitment, or lack thereof, to form gametocytes [40]. Hypnozoite populations at a similar time point post-infection were also thought to form 3 clusters corresponding to persisting, early activated, and late activated states [71]. Future work will be needed to determine the host-cell dependencies of these different parasite populations and to what extent parasite isolate influences their relative levels.

*Plasmodium* LS parasites are one of the fastest growing eukaryotes, yet very little is known about the intrinsic and environmental regulatory processes that control their size, although current data suggests that there are contributions from each. For example, host AMP-activated protein kinase (AMPK), which is likely inhibited by some drugs in our panel but is not part of the 300 kinases for which biochemical data is available [42], is linked to development of *P. berghei* and *P. falciparum* liver-stage size. Manipulation of host AMPK activity had an inverse effect on LS size. Reduction in parasite size corresponded to less parasite replication and lead to reduced total merosome numbers, not simply a delay in development [76]. Additionally, both host lipid uptake and regulation of host autophagic processes are required for optimal LS growth [30,76,77]. There is also strong evidence for parasite-intrinsic regulators of growth. For example, inhibition of both host Tak1 and *P. falciparum* protein kinase 9 (*Pf*PK9) increased the size of LS *P. berghei* [78]. Additionally, parasite fatty acid synthesis pathways are required for complete growth and development in rodent infectious species of *Plasmodium* [79].

The interplay between host signaling and parasite size is of particular interest for *P. vivax*, due to its long residence in the liver and its ability to switch from a non-dividing to a rapidly developing form. One possible hypothesis is that growth is biologically linked to dormancy, and the molecular machinery that underlies hypnozoite formation may also regulate schizont growth. This concept is not unprecedented in biology, as there are well-established linkages between cell size and cell division in other systems including bacteria [80], yeast [81], and several mammalian cell types [82,83]. Consistent with this hypothesis, we found that the majority of the few host kinases that were predicted to regulate hypnozoite levels across isolates were also predicted to regulate schizont size (Fig 8a). Specifically, the overlapping kinases were ERBB2, an oncogene [84], CAMK4, which regulates transcription, protein synthesis, and apoptosis and has a role hepatocellular carcinoma growth [85,86], and DYRK4, which regulates differentiation, apoptosis and sensitivity to DNA damage [87,88]. Investigating the link between dormancy and size, and how host regulators might influence parasite checkpoints in a way that is analogous to the established checkpoint model in other systems remains a critical area for future research.

While there is convincing evidence for parasite isolate determination of the ratio of schizont to hypnozoite formed upon infection [6,11,14], it remains an open question to what extent host conditions can skew this ratio. Our data support a role for host phosphosignaling in regulating the number of hypnozoites and schizonts present 8 dpi. We did not observe any kinase inhibitors that increased schizont while decreasing hypnozoite levels (Fig 3), which would have supported a model of host regulated conversion between hypnozoite and schizont forms in the initial period of infection. In all our experiments, drug treatment was added 24 hpi, when schizonts and hypnozoites are indistinguishable morphologically. We do not know at what point within the 7 days of treatment parasites are being cleared, or if there is a point in development when schizonts and hypnozoite forms diverge further in their sensitivity to host-targeted kinase inhibition. Future work utilizing a time course of treatment, as well as selective clearance of schizonts is warranted to further investigate the role of host phosphosignaling on hypnozoite formation and activation.

We observed only modest overlap in kinases predicted to regulate schizont numbers and size with our previously published KiR screen in *P. yoelii* [49]. In addition to parasite-intrinsic differences it is likely that differences in the host

environment between the primary hepatocyte and hepatoma cell lines used in the two studies may explain the absence of more overlap. The use of a primary or transformed cell line may be particularly relevant when considering kinases, such as ERBB2, that regulate cell growth and division [89]. ERBB2 was not predicted to regulate *P. yoelii* LS infection [49], though it has been predicted to regulate dengue virus infection [90] and KHSV reactivation [47].

Though host kinase inhibition was able to reduce the numbers of hypnozoites and schizonts, the variation in how different parasite isolates responded is a major concern when considering kinase inhibitors as potential candidates for host-targeted *P. vivax* therapeutics. Genetic and epigenetic variation between and within patient isolates are likely candidates for the variation seen here as pooling predictions from multiple isolates lead to the most robust predictions (**Figs 6**, **7**). Future work incorporating more data on parasite genetic variation may confirm or refute the existence of consistent host kinase regulators of *P. vivax* infection and their validity as therapeutic targets. We anticipate that as new systems become available for studying *P. vivax* LS infection and more screens are conducted, comparisons and modeling between forms and isolates, as was done here, will add depth to our understanding of the complexities of infection and the interactions between host and parasite.

## Study limitations

We attempted to minimize sources of variation by holding constant hepatocyte donor, CSP variant, province of isolate origin, and, for two of the three isolates, time of processing. However, we lacked additional information about the parasite isolates that could explain isolate variation, particularly the genetic heterogeneity between and within each sample [91,92]. We also do not have direct information on the infectivity of each isolate. We anticipate that differences in invasion rate would be captured in parasite counts 8 dpi, however we cannot deconvolve differences in invasion and infection maintenance in this study. We also cannot determine the replication dynamics that lead to the reduction in schizont size: whether this is due to early or late stalling of replication, or if the rate is steady but slower. The small number of parasite isolates available also limits our statistical power and ability to draw conclusions on the influence of isolate variation on susceptibility to host kinase inhibition.

Another limitation of this study is the inability to rule out off-target effects of inhibitors on parasite kinases. Inhibitors that influenced blood-stage infection with *P. falciparum* were removed from the screen, however, different parasite kinases may be expressed in *P. vivax* liver-stage infection that cannot be accounted for. Future work using genetic approaches will be needed to validate host kinase involvement in any phenotype.

Finally, the inhibitor screen described here only includes measured activity on 300 of the over 500 host kinases. When calculating the selective index for each phenotype, the effects of inhibitors on kinases that are not part of the screen are not incorporated. As substrate activity screens of these inhibitors expand in the future, the number of kinases that can be predicted will increase.

## Methods

### Ethics statement

All work was approved by the Ethics Committee of the Faculty of Tropical Medicine, Mahidol University (MUTM 2018-016-03).

### Generating *P. vivax* sporozoites

*Plasmodium vivax* infected blood was collected from patients attending malaria clinics in the Thasongyang district in Tak province, Thailand, under the protocol approved by the Ethics Committee of the Faculty of Tropical Medicine, Mahidol University (MUTM 2018-016-03). Written informed consent was obtained from each patient before sample collection. Infected blood was washed once with RPMI1640 incomplete medium before being resuspended with AB serum to a final 50% hematocrit and fed to female *Anopheles dirus* through membrane feeding. Engorged mosquitoes were maintained in

10% sugar solution until day 14–21 post feeding when salivary glands were dissected and sporozoites isolated. *P. vivax* mono-infection was confirmed by nested PCR. CSP variant was determined by RFLP-PCR.

### Infections

Primary human hepatocytes (lot ZPE, BioIVT) were plated in collagen-coated 384-well plates at 25,000 cells per well as described previously [53]. The same lot of cells was used for all experiments. Cells were maintained in InVitroGro CP medium (BioIVT) containing PennStrepNeo solution (Fisher Scientific) and Gentamicin (Fisher Scientific). A 3-well border of water was used around each plate to minimize edge effects. Two to three days post-plating, cells were infected with 14,000 hand-dissected *P. vivax* sporozoites per well. For each experiment multiple control wells were spread throughout the 384-well plate in case of any technical errors or contamination. In all experiments treatment began 24h post-infection. For initial KI screens all inhibitors were used at 500nM. For validation dose response curves, inhibitors were tested at several concentrations between 10nM to 10μM. In all cases media was changed and treatments refreshed every other day using a liquid handling robot (Opentrons). Cells were fixed 8 days post-infection using 4% PFA and then perme-abilized and blocked with 1% TritonX-100 in 2% BSA. Cells were stained with DAPI, diluted 1:2000, and an antibody against *Pv*UIS4 (gift from Dr. Noah Sather, Seattle Children's Research Institute), diluted 1:250, followed by anti-mouse AlexaFluor-594 (Invitrogen) diluted 1:1000 [56].

### Imaging and quantification

Plates were imaged using a Keyence BZ-X700 and quantified using ImageJ. Parasites were identified by positive PvUIS4 and DAPI staining and were categorized as schizonts (diameter > 10μm) or hypnozoites (diameter < 8μm, UIS4 promi-nence). To evaluate edge effects parasite levels were compared between outer (D5-15, L5-15) and inner (F5-15, J5-15) wells that received the same treatment. Parasite counts from all control wells were plotted, leading to uneven numbers between replicates. Schizont area was measured using PvUIS4 staining with ImageJ. Hepatocyte nuclei were counted using ImageJ as follows: DAPI channel image threshold was set with a pixel value lower limit of 69 and upper limit of 255. Nuclei were quantified using the measure particles feature within a size range of 40–800 pixel units and circularity between 0.4-1 to exclude debris and schizont DNA.

### Western blot

Primary human hepatocytes (lot ZPE) were plated as described above. Three days post-plating cells were treated with tucatinib (10nM – 10μM) or a DMSO control for 2 hours before stimulation with 10ng/mL recombinant human EGF (R&D Systems) for 10 minutes. Lysates were collected as described previously [93]. Gel electrophoresis was performed using Bolt 4–12% Bis-Tris mini protein gels. Proteins were transferred to PVDF membranes using iBlot 3 (Thermo Fisher Sci-entific Cat# A56727) dry blotting system. Blots were probed with primary antibodies against p-ERK1/2 (T202/Y204) (Cell Signaling Technology Cat# 4370) at 1:1000, and p-AKT (S473) (Cell Signaling Technology Cat# 9271) at 1:1000. Antibody to GAPDH (Cell Signaling Technology Cat# 97166) was used as loading control at 1:2000 dilution. Secondary antibodies rabbit IgG Horseradish Peroxidase-conjugated (R&D Systems Cat# HAF008) and mouse IgG Horseradish Peroxidase-conjugated (R&D Systems Cat# HAF007) were diluted 1:1000. Blots were imaged using Bio-Rad ChemiDoc imaging system and signals were quantified using ImageJ2 (https://imagej.nih.gov/ij/, version 2.3.0). Background correction was done for each band by subtracting background signals nearby the band.

### Sensitivity index

To create a sensitivity index (SI) for each phenotype a series of independent, random weights ranging from 0 and 1 were generated for the 34 kinase inhibitors in the screen. For each of the 300 kinases in the dataset, the sum of the weighted residual activity (catalytic activity upon kinase inhibitor treatment) from all 34 kinases inhibitors was calculated, after

which, standard deviation of the summed residual activity across the 300 kinases was calculated. The above procedure was repeated for 100,000 iterations, and the weights that led to the minimum standard deviation were assigned to the kinase inhibitors. The effect of each kinase inhibitor (percent change compared to DMSO) was then multiplied by the assigned weight. These weighted values were then added together to create the SI for each phenotype.

### Kinase Regression predictions

Kinase regression was performed independently for each parasite isolate and each phenotype, using the previously published algorithm described by Dankwa et al. [50]. Glmnet for Python (version 2.2.1) was used to fit the generalized linear models via penalized maximum likelihood, with the elastic net mixing parameter α of 0.8. The kinases with non-zero coefficients were predicted to be important for the phenotype of interest, and the effect of kinase inhibitors not tested in this study on that phenotype was also predicted based on the kinase predictions.

### Statistics

Distributions of all data sets were analyzed for normality using the Anderson-Darling and D'Agostino & Pearson tests. Parasite numbers and form ratio data were normally distributed and analyzed by ANOVA with Tukey's multiple comparisons test. Schizont size distributions were not normally distributed among all isolates and were analyzed using the Kruskal-Wallis test with Dunn's multiple comparisons test. To compare variances in schizont size between isolates, the data were ln(X) transformed before being analyzed by F test. Toxicity data were analyzed for all replicates by two-way ANOVA and Dunnett's multiple comparisons test. Paired t-tests were used to test for edge effects on infection. Inhibitor dose response data were analyzed by Fisher's LSD test. Pearson correlation coefficients were used to analyze all correlation data.

## Supporting information

**S1 Fig. Toxicity of kinase inhibitors in primary hepatocytes.** Primary hepatocyte nuclei were quantified in each infected well at the time of parasite quantification, 8 days post-infection, as a measure of kinase inhibitor toxicity. Each dot represents a technical replicate. N = 3–9. Data were normalized to the DMSO control average for each isolate and analyzed by Dunnett multiple comparisons test.
(TIF)

**S2 Fig. Well position did not influence parasite numbers.** (a) Schizont and (b) hypnozoite numbers were compared between outer edge (D5-15, L5-15) and inner (F5-15, J5-15) wells in each 384-well plate (one plate per isolate) that received the same treatment. Each dot represents counts from a single well. Data were analyzed by paired t-test.
(TIF)

**S3 Fig. Promiscuous inhibitors do not show greater inhibition of infection.** The effect of each inhibitor on schizont or hypnozoite numbers, for isolate A, is plotted against the number of kinases inhibited by each inhibitor using cut-offs of 20%, 50% or 80%. Each dot represents a single inhibitor. Pearson correlation coefficients (r) and p-values (p) were calculated for each plot.
(TIF)

**S4 Fig. Schizont sensitivity to inhibitors varies across isolates.** (a) Correlation between effect of kinase inhibitors on the number of schizonts, normalized to DMSO controls, in isolate A vs isolate B. Gray shaded area indicates a 95% confidence interval, based on the variation of control wells, around x = y. Each dot represents an inhibitor. Inhibitors for which dose response curves were done are highlighted. (b) Schizont dose response curves for two kinase inhibitors. Each dot represents a technical replicate. Data were analyzed by Fisher's LSD test.
(TIF)

**S5 Fig. Correlation of effect on size between isolates and with infection.** (a) Schizont area, in pixels, plotted against raw internal density (RawIntDen) of DAPI signal within the parasite in control wells. Each dot represents a single parasite from a DMSO-treated well. (b) The mean effect of kinase inhibitors on schizont size plotted for each pair of parasite isolates. Each dot represents a kinase inhibitor. (c) Mean effect of kinase inhibitors on parasite size plotted against mean effect on number of schizonts for each isolate. Each dot represents an inhibitor. Pearson correlation coefficients (r) and p-values (p) were calculated for each comparison.
(TIF)

**S6 Fig. Tucatinib inhibits phospho-signaling downstream of ERBB2.** Images (a) and quantification (b) of western blots measuring ERK1/2 and AKT phosphorylation in lysates from uninfected primary human hepatocytes 10 minutes post EGF (10ng/mL) stimulation, with or without a 2 hour tucatinib (10nM-10μM) pre-treatment. Data were normalized to GAPDH loading controls and to the no-treatment (NT) condition. (c) Number of schizonts, (d) schizont size, and (e) number of hypnozoites 5 dpi in parasite isolate E in response to tucatinib treatment (100nM-10μM). Treatment was begun 24 hpi and maintained until fixation. Each dot represents a technical replicate (c, e) or a single parasite (d). Data were analyzed by Kruskal-Wallis test with Dunn's multiple comparisons test.
(TIF)

**S1 Table. Related to** Fig 2. **Panel of kinase inhibitors used to treat** *P. vivax*-**infected hepatocytes.**
(XLSX)

**S2 Table. Related to** Fig 7. **Kinases and kinase inhibitors predicted to regulate schizont levels, hypnozoite levels, and schizont size for each parasite isolate.**
(XLSX)

**S3 Table. Related to** Fig 4b, 4c. Correlation between predicted and tested effects of predicted kinase inhibitors in isolate B.
(DOCX)

**S4 Table. Related to** Fig 6a, 6b. Predicted schizont and hypnozoite rates, compared to DMSO controls, in isolates A, B, and C for kinase inhibitors tested in Fig 6a, 6b.
(DOCX)

**S1 File. Raw data for all Figs.**
(XLSX)

## Acknowledgments

We thank Dennis Kyle and Steven Maher at University of Georgia for helpful conversations on primary hepatocyte and *P. vivax* culture.

## Author contributions

**Conceptualization:** Elizabeth KK Glennon, Alexis Kaushansky.

**Formal analysis:** Elizabeth KK Glennon, Ling Wei.

**Funding acquisition:** Elizabeth KK Glennon, Alexis Kaushansky.

**Investigation:** Elizabeth KK Glennon, Wanlapa Roobsoong, Veronica I Primavera, Elizabeth M van Zyl, Tinotenda Tongogara, Conrad B Yee.

**Methodology:** Ling Wei.

**Resources:** Wanlapa Roobsoong, Jetsumon Sattabongkot.

**Supervision:** Jetsumon Sattabongkot, Alexis Kaushansky.

**Writing – original draft:** Elizabeth KK Glennon.

**Writing – review & editing:** Elizabeth KK Glennon, Ling Wei, Wanlapa Roobsoong, Veronica I Primavera, Elizabeth M van Zyl, Tinotenda Tongogara, Conrad B Yee, Jetsumon Sattabongkot, Alexis Kaushansky.

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
