## [Decision Letter · Decision Letter 0]

10 Jun 2025

Host kinase regulation of Plasmodium vivax dormant and replicating liver stages

Thank you for submitting your manuscript to PLOS Neglected Tropical Diseases. After careful consideration, we feel that it has merit but does not fully meet PLOS Neglected Tropical Diseases's publication criteria as it currently stands. Therefore, we invite you to submit a revised version of the manuscript that addresses the points raised during the review process.

Please submit your revised manuscript within 30 days Aug 09 2025 11:59PM. If you will need more time than this to complete your revisions, please reply to this message or contact the journal office at plosntds@plos.org. Please include the following items when submitting your revised manuscript:

* A rebuttal letter that responds to each point raised by the editor and reviewer(s). You should upload this letter as a separate file labeled 'Response to Reviewers '. This file does not need to include responses to any formatting updates and technical items listed in the 'Journal Requirements' section below.

* A marked-up copy of your manuscript that highlights changes made to the original version. You should upload this as a separate file labeled 'Revised Manuscript with Track Changes '.

* An unmarked version of your revised paper without tracked changes. You should upload this as a separate file labeled 'Manuscript '.

We look forward to receiving your revised manuscript.

Kind regards,

Luther A Bartelt

Academic Editor

Sarman Singh

Section Editor

Shaden Kamhawi

co-Editor-in-Chief

Paul Brindley

co-Editor-in-Chief

**Journal Requirements:**

At this stage, the following Authors/Authors require contributions: Elizabeth M van Zyl. Please ensure that the full contributions of each author are acknowledged in the "Add/Edit/Remove Authors" section of our submission form.

2) We note that your Figure S6 - new, and Figure S6 files are duplicated on your submission. Please remove any unnecessary or old files from your revision, and make sure that only those relevant to the current version of the manuscript are included.

3) Your manuscript is missing the following sections: Abstract.  Please ensure all required sections are present and in the correct order. Make sure section heading levels are clearly indicated in the manuscript text, and limit sub-sections to 3 heading levels. An outline of the required sections can be consulted in our submission guidelines here:

Potential Copyright Issues:

i) Figure 4A. Please confirm whether you drew the images / clip-art within the figure panels by hand. If you did not draw the images, please provide (a) a link to the source of the images or icons and their license / terms of use; or (b) written permission from the copyright holder to publish the images or icons under our CC BY 4.0 license. Alternatively, you may replace the images with open source alternatives. See these open source resources you may use to replace images / clip-art:

5) We note that your Data Availability Statement is currently as follows: "All data used in this submission are within the manuscript and supplemental files.". Please confirm at this time whether or not your submission contains all raw data required to replicate the results of your study. Authors must share the “minimal data set” for their submission. PLOS defines the minimal data set to consist of the data required to replicate all study findings reported in the article, as well as related metadata and methods (https://journals.plos.org/plosone/s/data-availability#loc-minimal-data-set-definition).

6) Please ensure that the funders and grant numbers match between the Financial Disclosure field and the Funding Information tab in your submission form. Note that the funders must be provided in the same order in both places as well.

**Reviewers' comments:**

Reviewer's Responses to Questions

**Key Review Criteria Required for Acceptance?**

**Methods**

-Are the objectives of the study clearly articulated with a clear testable hypothesis stated?

-Is the study design appropriate to address the stated objectives?

-Is the population clearly described and appropriate for the hypothesis being tested?

-Is the sample size sufficient to ensure adequate power to address the hypothesis being tested?

-Were correct statistical analysis used to support conclusions?

-Are there concerns about ethical or regulatory requirements being met?

Reviewer #1: -Are the objectives of the study clearly articulated with a clear testable hypothesis stated?

yes

-Is the study design appropriate to address the stated objectives?

Positive control are missing when addressing KI effect on the schizonts. I think it would be important to explain why the number of repetition (wells) isn’t the same for each conditions of figure 1. Authors should indicate if sporozoite viability have been tested or not

-Is the population clearly described and appropriate for the hypothesis being tested?

Yes

-Is the sample size sufficient to ensure adequate power to address the hypothesis being tested

yes

-Were correct statistical analysis used to support conclusions?

yes

-Are there concerns about ethical or regulatory requirements being met?

I haven’t noticed ethical concerns

Reviewer #2: Yes

**Results**

-Does the analysis presented match the analysis plan?

-Are the results clearly and completely presented?

-Are the figures (Tables, Images) of sufficient quality for clarity?

Reviewer #1: -Does the analysis presented match the analysis plan?

yes

-Are the results clearly and completely presented?

yes

-Are the figures (Tables, Images) of sufficient quality for clarity?

yes

Reviewer #2: Yes

**Conclusions**

-Are the conclusions supported by the data presented?

-Are the limitations of analysis clearly described?

-Do the authors discuss how these data can be helpful to advance our understanding of the topic under study?

-Is public health relevance addressed?

Reviewer #1: -Are the conclusions supported by the data presented?

yes

-Are the limitations of analysis clearly described?

Yes

-Do the authors discuss how these data can be helpful to advance our understanding of the topic under study?

yes

-Is public health relevance addressed?

yes

Reviewer #2: Yes

**Editorial and Data Presentation Modifications?**

Reviewer #1: L183 perhaps recalcitrant isn’t the most accurate term, I would keep it simple an say “not susceptible” or similar.

Reviewer #2: (No Response)

**Summary and General Comments**

Reviewer #1: Glennon et al. investigated the dependencies of P. vivax liver stages (schizont, hypnozoite) to host phosphosignaling pathways through evaluation of kinase inhibitors. To carry out their investigations the authors used a LS culture system based on human primary hepatocytes and imaging post UIS4 marking (fluorophore coupled) as readout. This methodology was proved to be adequate in multiple studies. Unfortunately, these results are based on a limited number of isolates but this has to be put in perspective with the low P. vivax incidence today in south east Asia and the demanding aspect of Pv LS culture. The authors also present limitations, I essentially agree with, of their study.

Results

Authors presented first results of hypnozoite production rate and number of parasites. These variables vary in a certain extent according to the isolate of origin. This finding is not particularly innovative. In addition, the variable number of well analyzed per isolates, the large range of value obtained is, to me, making conclusions out these experiments a bit uncertain (see results comments) even if those are to me consistent with the literature. There is from my perspective a limited added value to have these results presented here, perhaps SI section would be more suitable. Then the authors present results based on host kinase inhibition vs. hypnozoite and schizont numeration. ¾ isolates collected were used to do so. The results about how some specific host KI might influence the development of the parasite is promising although deeper investigation appear necessary. The conclusion is based on an association and extensive further biochemical measurement would be necessary. Using these results, the authors modelized the activity of a larger pool of KI reaching 178 potential drugs presenting an effect. Out of these 178 drugs, 5 were to my understating actually verified in the LS model. The conclusion is a certain weakness of KIR model to predict the activity with an attributed major impact of strain variability unlike data get pooled. Despite the interest of this part, I’ve found the absence of specific schizont inhibitor control very detrimental. Finally, the authors concertized their investigation on host kinase through evaluating the impact of ERBB2 inhibition, previously identified as potentially favoring parasites development. Impact on schizont looks clear and dose dependent unlike what has been observed with hypnozoite. Here again a positive control would have been welcome.

L117 Why the number of sporozoites isolated out of isolate A isn’t mentioned?

Fig1b/c, it is not clear to me why isolates data appeared to be based on a different number of wells? Higher wells numbers ie in A and D gave higher value spread (which is normal) then it don’t really sounds to me fully comparable with B and C as higher wells number might change the conclusion.

L182 the observation of concomitant reduction of schizont and hypnozoite numbers unlike data generated by other groups with specific schizonticides can also be indeed a matter of specificity and might also be the reflect of the inherent toxicity/metabolism alteration of the KI to the host cells.

Discussion

L325: the data presented are not consistent with the variable reactivation temporality observed in the different geographic settings. Authors should rephrase by mentioning only the infection rate aspect.

L434 The “origin” of a given P. vivax isolate ie relapse or de novo infection is indeed particularly complex to determine but does this represent a variation factor more than parasites population heterogenicity? To reinforce this statement a citation would be welcome.

The authors should also discuss the drugging timing, in the present design they saw variable activity by adding drug at 24h PI. Effect or not on later parasites should be discussed.

Methodology

CSP typing is presented in the manuscript as limited (I agree with this statement) so finally why using such approach especially when non-confidential approaches such as whole genome sequencing or targeted amplicon sequencing exist and would be more informative.

For the infection the authors standardize their inoculum to 14K sporozoites/well but does sporozoites viability for each batch has been verified?

Reviewer #2: To learn whether responses to host (human) kinases govern schizont/hypnozoite growth (and therefore relapse patterns) in vivax parasites, Glennon and colleagues use 4 P. vivax isolates from NW Thailand of the same CSP variant to infect liver cells from the same primary donor and attempt to discover kinases that alter parasite schizont and hypnozoite numbers and sizes in the same way across isolates. Unfortunately, variation in infection phenotype and responses to kinases between isolates makes it hard to pinpoint any common host kinase pathways or kinase inhibitors that consistently/predictably affect the parasite phenotypes investigated. In particular, the authors were limited to two robust isolates (A&B), with C infecting low numbers of cells, and isolate D saved for replication experiments, which as a whole, did not yield robust conclusions or targets. The authors were forced to test inhibitors that acted similarly in 2 of 3 isolates. Since the six kinase inhibitors tested for validation all reduced schizont size at the highest concentration, I am not sure if the prior work was indeed predictive, or if many inhibitors are just toxic at high concentration.

The authors conclude that better characterization of the genetic diversity of isolates used (and therefore ability to select genetically similar isolates to probe) may help reduce the variability in their outcomes. It is true that even a single “isolate” with the same broad CSP variant identified by PCR (RFLP) might actually contain different parasite variants (“strains”) arising from separate hypnozoites and this within-host genetic diversity could lead to a spectrum of phenotypic outcomes even within the same isolate. Also, it is hard to ignore the extent of global genetic diversity of P vivax (S American vs South Asian vs SE Asian isolates), making even robust outcomes in a genetically homogenous groups of isolates hard to translate more widely. I agree with a prior reviewer that the work is a good starting point, and hopefully the authors can further investigate their subset of kinases in a (much) larger, more diverse set of clinical isolates (that includes subsets of genetically similar isolates), while also thinking about the effects of kinase inhibitor bioavailability and epigenetic variation on their experiments.

Though I have no firsthand experience with this type of experimental work, I agree with the prior reviewers that the experiments involve high technical difficulty, and at least in the revised version, the authors go through pains to clearly explain their results and next steps through multiple figures and tables. In response to previous reviewers, the authors have provided further validation of their conclusions by using tucatinib to test the effect of ERBB2 inhibition (as well as 5 other kinase inhibitors) on schizont/hypnozoite numbers and size, with some predictable effect on these phenotypes - mostly reducing schizont numbers in a dose response patterns, in two different isolates.

The paper underwent a thorough review at PLoS Pathogens, and the authors appear to have taken a rigorous approach to addressing those previous comments and feedback.

Minor comments:

Line 72 – “blocking apoptosis” � blocking of apoptosis

Line 213 – “only two of the five inhibitors, SB203580 and JNK inhibitor V, significantly reduced schizont numbers in isolate B at any of the concentrations tested (Fig. 4c)”

In Fig 4C, JNK inhibitor V reduced schizont numbers at 10uM, but increased schizont numbers at 1uM. It is odd to single this out given the dose response design. I might only highlight the first inhibitor which achieves a clear dose reponse effect. PKR inhib looks look it might show a dose response effect if more replicates were done.

On a related note, the effects noted in Fig 6 all occur only at 10uM, which the authors might comment on in their discussion.

PLOS authors have the option to publish the peer review history of their article (what does this mean? ). If published, this will include your full peer review and any attached files.

**Do you want your identity to be public for this peer review?** For information about this choice, including consent withdrawal, please see our Privacy Policy .

Reviewer #1: No

Reviewer #2: No

**Figure resubmission:**
---

## [Decision Letter · Decision Letter 1]

12 Nov 2025

Response to Reviewers
Revised Manuscript with Track Changes
Manuscript

Shaden Kamhawi

co-Editor-in-Chief

Paul Brindley

co-Editor-in-Chief

**Additional Editor Comments :**

We have received your revision, and the reviewers indicate that multiple prior concerns have been addressed. There are, however, some additional requests prior to making a final decision. Please see requests for clarification of the controls and analytical choices in the presented work.

**Journal Requirements:**

1) We have noticed that you have a list of Supporting Information legends in your manuscript for S1 Figures to S5 Figures. However, there are no corresponding files uploaded to the submission. Please upload them as separate files with the item type 'Supporting Information'.

**Reviewers' comments:**

Reviewer's Responses to Questions

**Key Review Criteria Required for Acceptance?**

**Methods**

-Are the objectives of the study clearly articulated with a clear testable hypothesis stated?

-Is the study design appropriate to address the stated objectives?

-Is the population clearly described and appropriate for the hypothesis being tested?

-Is the sample size sufficient to ensure adequate power to address the hypothesis being tested?

-Were correct statistical analysis used to support conclusions?

-Are there concerns about ethical or regulatory requirements being met?

Reviewer #1: yes

Reviewer #3: Generally, the methods are sufficient to suppor the conclusions of the methods and hypothesis being tested. However, there are a few outstanding questions listed below that would help provide clarity on the choices the authors made during the analysis.

**Results**

-Does the analysis presented match the analysis plan?

-Are the results clearly and completely presented?

-Are the figures (Tables, Images) of sufficient quality for clarity?

Reviewer #1: yes

Reviewer #3: yes

**Conclusions**

-Are the conclusions supported by the data presented?

-Are the limitations of analysis clearly described?

-Do the authors discuss how these data can be helpful to advance our understanding of the topic under study?

-Is public health relevance addressed?

Reviewer #1: yes

Reviewer #3: yes

**Editorial and Data Presentation Modifications?**

Reviewer #1: i have no further modifications to suggest.

Reviewer #3: None

**Summary and General Comments**

Reviewer #1: I would like to thanks the authors for having addessed all my suggestions/points.

Reviewer #3: Glennon et al. present an investigation into the role of host phosphosignaling during Plasmodium vivax liver stage infect, aiming to elucidate how these pathways influence parasite development, including hypnozoite formation and schizont development. The manuscript provides evidence that host hepatocyte kinase signaling contributes to liver stage biology, while acknowledging several limitations that are appropriately discussed. The authors have addressed prior reviewer comments effectively; however, the following points require clarification before acceptance.

1. Inclusion of MMV390048 Data- Incorporating MMV390048 data alongside the current dataset would enhance interpretability. While concerns about duplicating data are valid, proper citation and clear indication in the results should mitigate this issue. Including these data would allow readers to more easily assess hypnozoite and schizont observations and help with navigating the interpretations being drawn.

2. Measurement Approach - Please clarify the rationale for using parasite diameter rather than area in the analysis of P. vivax liver stages to classify hypnozoites versus schizonts. Generally, area is used.

3. VK210 Variant - The manuscript notes parasite-intrinsic factors; however, it would be valuable to discuss how these findings confirm or challenge previous data on VK210, given that all parasites studied are VK210 variants.

4.Assessment of Hepatocyte Viability - The authors state that toxic compounds were removed, but was hepatocyte viability assessed (e.g., via hepatocyte nuclei counts)? If performed, please indicate where this is reported; if not, consider adding this analysis to confirm that toxicity does not confound the results.

5. Interpretation of Figure 2- How do the authors reconcile slowed schizont growth with changes in hypnozoite numbers? Could some parasites classified as hypnozoites represent slowed schizonts? Addressing this possibility earlier would help bridge gaps left by missing controls noted by other reviewers.

6. The statement “we observe a strong positive correlation between the impact of a given inhibitor on schizonts and hypnozoites…” appears to suggest hypnozoites are unaffected by kinase manipulation, yet other sections imply otherwise. Please clarify this point. Additionally, since hypnozoites increase in size between days 3 and 8, was size shifting analyzed within this population?

7. Normalization in Figure 4 -- Why were parasite counts presented per well rather than normalized to controls with known outcomes on Pv liver stages? For example, normalization to monensin or nigericin (which eliminate all parasite forms) would add rigor compared to raw counts.

8.The manuscript states that the data explain 'heterogeneous outcomes observed with P. vivax isolates', but how the data actually contribute to this is unclear. The results primarily indicate that in vitro perturbations occur, likely influenced by cell type choices, as noted by the authors. Please clarify how the presented data support this conclusion.

9. When kinase treatment strongly correlated with schizont inhibition, did it also affect hypnozoite size, or was growth slowing exclusive to schizonts?

PLOS authors have the option to publish the peer review history of their article (what does this mean? ). If published, this will include your full peer review and any attached files.

**Do you want your identity to be public for this peer review?** For information about this choice, including consent withdrawal, please see our Privacy Policy .

Reviewer #1: No

Reviewer #3: No

**Figure resubmission:****Reproducibility:** To enhance the reproducibility of your results, we recommend that authors of applicable studies deposit laboratory protocols in protocols.io, where a protocol can be assigned its own identifier (DOI) such that it can be cited independently in the future. Additionally, PLOS ONE offers an option to publish peer-reviewed clinical study protocols. Read more information on sharing protocols at https://plos.org/protocols?utm_medium=editorial-email&utm_source=authorletters&utm_campaign=protocols

---

## [Decision Letter · Decision Letter 2]

18 Feb 2026

Dear Dr. Kaushansky,

We are pleased to inform you that your manuscript 'Host kinase regulation of Plasmodium vivax dormant and replicating liver stages' has been provisionally accepted for publication in PLOS Neglected Tropical Diseases.

Best regards,

Luther A Bartelt

Academic Editor

Sarman Singh

Section Editor

Shaden Kamhawi

co-Editor-in-Chief

Paul Brindley

co-Editor-in-Chief

Reviewer's Responses to Questions

**Key Review Criteria Required for Acceptance?**

**Methods**

-Are the objectives of the study clearly articulated with a clear testable hypothesis stated?

-Is the study design appropriate to address the stated objectives?

-Is the population clearly described and appropriate for the hypothesis being tested?

-Is the sample size sufficient to ensure adequate power to address the hypothesis being tested?

-Were correct statistical analysis used to support conclusions?

-Are there concerns about ethical or regulatory requirements being met?

Reviewer #3: (No Response)

**Results**

-Does the analysis presented match the analysis plan?

-Are the results clearly and completely presented?

-Are the figures (Tables, Images) of sufficient quality for clarity?

Reviewer #3: (No Response)

**Conclusions**

-Are the conclusions supported by the data presented?

-Are the limitations of analysis clearly described?

-Do the authors discuss how these data can be helpful to advance our understanding of the topic under study?

-Is public health relevance addressed?

Reviewer #3: (No Response)

**Editorial and Data Presentation Modifications?**

Reviewer #3: (No Response)

**Summary and General Comments**

Reviewer #3: (No Response)

PLOS authors have the option to publish the peer review history of their article (what does this mean? ). If published, this will include your full peer review and any attached files.

**Do you want your identity to be public for this peer review?** For information about this choice, including consent withdrawal, please see our Privacy Policy .

Reviewer #3: No

---

## [Editor Report · Acceptance letter]

Dear Dr. Kaushansky,

We are delighted to inform you that your manuscript, "Host kinase regulation of Plasmodium vivax dormant and replicating liver stages," has been formally accepted for publication in PLOS Neglected Tropical Diseases.

Best regards,

Shaden Kamhawi

co-Editor-in-Chief

Paul Brindley

co-Editor-in-Chief
